

# A survey of *Cryptosporidium* prevalence among birds in two zoos in China

Yaxian Lu[1,2], Tianchun Pu[3], Baohua Ma[1,2], Lixin Wang[1,2], Mengchao Zhou[1,2], Yu Chen[1,2], Xiuyun Li[4], Changming Zheng[3], Hetong Liu[1,2], Jinpeng Liu[3], Chunyu Guan[4], Hongyan Yu[1,2], Chunkuo Dai[3], Yuan Huang[1,2], Yuling Yang[1,2], Zhiwei Peng[1,2], Lei Han[1,2], Hongliang Chai[1] and Zhijun Hou[1,2]

[1] Northeast Forest University, Harbin, China
[2] Laboratory of Vector-Borne Diseases and Pathogens Ecology, Northeast Forest University, Harbin, China
[3] Beijing Key Laboratory of Captive Wildlife Technologies, Beijing, China
[4] Harbin North Forest Zoo, Harbin, China

Corresponding authors
Hongliang Chai,
Hongliang_chai@hotmail.com
Zhijun Hou, houzhijundb@163.com

## ABSTRACT

**Background**. Cryptosporidiosis is an important zoonotic protozoan disease worldwide, but few studies on this disease have been performed in wild birds; thus, our knowledge of this disease is insufficient, even in zoo birds. Animals in zoos are possible zoonotic disease reservoirs, potentially resulting in zoonotic agent spillover to humans; accordingly, our understanding of such phenomena should be improved.

**Methods**. A total of 263 fresh fecal samples from 43 avian species were randomly collected from the Beijing Zoo and Harbin North Forest Zoo and screened for the prevalence of *Cryptosporidium* by 18S rRNA gene sequencing. *Cryptosporidium* species were distinguished based on the combined results of phylogenetic tree and genetic distance analyses conducted with the inclusion of seven avian *Cryptosporidium* species and 13 avian *Cryptosporidium* genotypes. The genetic diversity of *Cryptosporidium parvum* among different hosts, including humans, cattle, dogs, and birds, and the genetic diversity of avian *C. parvum* among avian hosts in China, Iraq and Brazil were determined based on *C. parvum* 18S rRNA haplotypes.

**Results**. The results of PCR targeting the 18S rRNA gene revealed that 1.9% (5/263) of the samples were *Cryptosporidium*-positive. Four of the five *Cryptosporidium*-positive samples originated from white cranes (*Grus leucogeranus*), and one originated from a flamingo (*Phoenicopteridae*). Avian *C. parvum* isolates, including the isolates examined in the present study, showed gene flow with other isolates from different types of hosts, including humans, cattle and dogs, indicating that zoo birds potentially pose zoonotic and pathogenic risks to humans and animals. Additionally, gene flow between avian *C. parvum* isolates from China and Brazil was detected.

**Conclusions**. To the best of our knowledge, our results demonstrate *C. parvum* infection in a flamingo (*Phoenicopteridae*) and white cranes (*Grus leucogeranus*) for the first time. The results of our study provide an important reference for understanding the host range, biological characteristics, and molecular epidemiology of *C. parvum*.

## INTRODUCTION

*Cryptosporidium* infections are prevalent in fish, amphibians, reptiles, birds and mammals worldwide (*Nakamura & Meireles, 2015*; *Ryan, 2010*), and *Cryptosporidium* is an important pathogen that causes diarrhea in its hosts, especially in people infected with HIV (*Caccio & Chalmers, 2016*; *Wang et al., 2018*). The first discovery and description of *Cryptosporidium* in birds were reported by Tyzzer (*El-Sherry et al., 2014*) in 1929. To date, seven valid *Cryptosporidium* species, including *Cryptosporidium meleagridis*, *Cryptosporidium parvum*, *Cryptosporidium galli*, *Cryptosporidium baileyi*, *Cryptosporidium avium*, *Cryptosporidium muris*, and *Cryptosporidium andersoni* and specific *Cryptosporidium* genotypes, including avian genotypes I–IV; and VI–IX; goose genotypes I–IV; duck genotype; Eurasian woodcock genotype; finch genotype; ostrich genotype and *Cryptosporidium* xiaoi-like genotype have been documented in birds worldwide (*Plutzer & Karanis, 2009*). *Cryptosporidium* has been reported in more than 30 avian hosts belonging to Anseriformes, Charadriiformes, Columbiformes, Galliformes, Passeriformes, Psittaciformes, Falconiformes, Gruiformes, Ciconiiformes, Neognathae, Bucerotiformes and Struthioniformes worldwide (*Azmanis et al., 2018*; *Dong et al., 2021*; *Ferrari et al., 2018*; *Gajadhar, 1994*; *Graczyk et al., 1998*; *Jellison et al., 2004*; *Jian et al., 2021*; *Kabir et al., 2020*; *Majewska et al., 2009*; *Ng, Pavlasek & Ryan, 2006*; *Oliveira et al., 2017*; *Plutzer & Tomor, 2009*; *Quah et al., 2011*; *Wang et al., 2019*; *Zylan et al., 2008*).

*C. parvum* infection is an important zoonotic protozoan disease that can affect humans, dogs, alpaca, cattle, horses, and turtles (*Abe, Kimata & Iseki, 2002*; *Couso-Perez et al., 2020*; *Gomez-Puerta et al., 2020*; *Rzezutka, Kaupke & Gorzkowski, 2020*). *C. parvum* infection is also prevalent in avian species. *C. parvum* can infect chicken, pigeons, parrots, Bengalese finches, stone curlew, mandarin ducks, carrion crows and captive falcons among poultry species (*Azmanis et al., 2018*; *Ferrari et al., 2018*; *Kabir et al., 2020*; *Majewska et al., 2009*; *Oliveira et al., 2017*; *Zylan et al., 2008*); Canada geese, mallards, coots, whooper swans, common mergansers, mute swans, rook and white storks among wild birds (*Graczyk et al., 1998*; *Majewska et al., 2009*; *Plutzer & Tomor, 2009*; *Wang et al., 2019*), and black swans, crestless fireback pheasants, Fischer's lovebirds, golden pheasants, great curassows, pink backed pelicans and wrinkled hornbills among captive species in the National Zoo of Malaysia in Kuala Lumpur (*Quah et al., 2011*). However, reports of *Cryptosporidium* infection in avian species are scarce in comparison to those in mammals.

The small intestine and cecum will develop lesions when avian are infected with *C. parvum* (*Nakamura & Meireles, 2015*). An outbreak of *C. parvum* infection among stone curlews was reported in Dubai; the symptoms included sudden-onset diarrhea, lethargy and reduced feed intake (*Zylan et al., 2008*). However, detailed descriptions of wild birds infected with *C. parvum* are lacking.

Species identification is a crucial step in preventing and curing *Cryptosporidium* infections, but it is difficult to distinguish species based on only morphology, as they share similar morphological characteristics (*Ryan, Fayer & Xiao, 2014*). Molecular methods targeting markers in the 18S rRNA gene are useful, practical tools for *Cryptosporidium* species identification, with high detection limits (*Ryan et al., 2003*).
Zoos are potential environmental sources of zoonotic diseases because they serve as temporary homes for wild birds. Tourists visiting a zoo may come into close contact with certain wild birds, which is a practical pathway of *C. parvum* spillover from birds to humans. Therefore, this pathway should be studied to increase our understanding of these spillover events. The objectives of the present study were (1) to investigate *Cryptosporidium* species and their prevalence among zoo birds in China and (2) to evaluate the potential risk of transmission of *Cryptosporidium* from zoo birds to other animals or humans.

## MATERIALS & METHODS

### Fecal samples

We commissioned the zoo staff to help us collect samples, all staff wore disposable gloves and placed bird droppings into sterile centrifuge tubes. All samples were transported to the laboratory on dry ice. In the laboratory, the samples were stored in a sterile centrifuge tube containing a 5% potassium dichromate solution in a refrigerator at 4 °C until analysis. A total of 263 fresh fecal samples from birds belonging to 16 families, 30 genera, and 43 species were collected from March 2019 to September 2019 at the Beijing Zoo and the Harbin North Forest Zoo (Table S1). Among the 263 fecal samples, 66 were from the Harbin Zoo, and 197 were from the Beijing Zoo. Parasite eggs were separated in a saturated solution of sucrose used as floating medium and stored at −20 °C until DNA extraction.

### Molecular analyses

DNA was extracted directly from the samples by using the Stool DNA Kit (Omega Bio-Tek, China Technical Support Center) following the manufacturer's instructions. Nucleic acids were eluted in 50 µl of elution buffer to increase the quantity of DNA recovered. The 18S rRNA gene sequence was amplified by nested PCR (*Koehler et al., 2018*). The primer sets 18SCiF2: 5′-GACATATCATTCAAGTTTCTGACC-3′ and 18SCiR2: 5′-CTGAAGGAGTAAGGAACAACC-3′ were selected for the primary reaction, and 18SCiF1: 5′-CCTATCAGCTTTAGACGGTAGG-3′ and 18SCiR1: 5′-TCTAAGAATTTCACCTCTGACTG-3′ were employed for the secondary reaction. The reaction mixture consisted of 11 µl of double distilled water, 2 µl of each primer (Comate Biosciences Co., Ltd. Changchun, China), 25 µl of Taq DNA polymerase (TaKaRa, Dalian, China), and 10 µl of template DNA, with a total reaction volume of 50 µl. The PCR program consisted of an initial heating step at 94 °C for 5 min, followed by 45 cycles of 94 °C for 30 s, 55 °C for 30 s, and 72 °C for 30 s (for primary and nested PCR) and a final extension step of 7 min at 72 °C. Each PCR product was identified in a 0.8% agarose gel to verify its size. The PCR products were sent to Comate Biosciences Co., Ltd. (Changchun, China), for sequencing.

The 18S rRNA gene fragment sequences of *Cryptosporidium* examined in the present research were aligned using DNAMAN (v6.0, Lynnon BioSoft, Quebec, Canada) and submitted to GenBank under accession numbers MW664001–MW664003 (G3, G5, G6, host: white crane), MW664005 (P1, host: flamingo), and MW664006 (G4, host: white crane).

Phylogenetic trees were constructed by using the MEGA 7.0 program based on evolutionary distances *via* Maximum Likelihood (ML) analysis, with the Tamura 3-parameter model for the 18S rRNA genes of 7 avian *Cryptosporidium* species (*C. meleagridis*, *C. parvum*, *C. galli*, *C. baileyi*, *C. avium*, *C. muris* and *C. andersoni*) and 13 avian *Cryptosporidium* genotypes (avian genotypes I–III, and VI–IX; goose genotypes I–II; duck genotype; finch genotype; ostrich genotype; and C. xiaoi-like genotype), with 1,000 bootstrap replicates. The sequence of *Sarcocystis tenella* (MH413042) was selected as the outgroup.

The haplotypes of the 18S rRNA genes of 7 avian *Cryptosporidium* species and *C. parvum* from 4 different host classes were analyzed by using DnaSP 5.0 software. A phylogenetic tree of the *Cryptosporidium* spp. haplotypes was constructed with the same parameters mentioned above. The genetic distances between different *Cryptosporidium* species were calculated with the MEGA 7.0 program. Arlequin 3.0 was used to calculate the pairwise fixation index (Fst) values of *C. parvum* between different host classes and different countries using the 18S rRNA gene. The migration number of each generation (Nm) was determined according to the following formula: Fst = 1/ (4Nm + 1). The corresponding heatmap was generated with the TBtools program. We used statistical and descriptive analysis in infection rate calculation.

## RESULTS

Five fecal samples from four white cranes (*Grus leucogeranus* G3, G4, G5, G6) and one flamingo (*Phoenicopteridae* P1) were *Cryptosporidium*-positive. The positive rate for *Cryptosporidium* was 1.9% (5/263). *Cryptosporidium* isolates from five positive fecal samples were most closely related to *C. parvum* (Fig. 1).

We found 5, 3, 1, 6, 1, 3 and 8 unique 18S rRNA gene haplotypes in avian *C. parvum*, *C. meleagridis*, *C. avium*, *C. baileyi*, *C. muris*, *C. andersoni* and *C. galli*, respectively. Figure 2 shows that the *Cryptosporidium*-positive samples (G3, G4, G5, G6 and P1) were included in the clade of *C. parvum*, which was parallel to the clade of *C. meleagridis*. *C. parvum* Hap1 and *C. parvum* Hap 5 were defined in the present study. Hap1 was shared between avian *C. parvum* from white crane and flamingo in the present study and has been reported previously in other species. However, Hap5 was found in only white cranes (Table S2).

The analysis of the genetic distances between the 31 haplotypes of the 18S rRNA genes in the 7 avian *Cryptosporidium* species showed that the intraspecific genetic distances of *C. parvum* were within 0.007, but the interspecific genetic distances between *C. parvum* and the other six *Cryptosporidium* species were all greater than 0.007 (Fig. 2), which together with the results of the phylogenetic tree analysis, verified that the *Cryptosporidium* species obtained in the present study was *C. parvum*.

We identified 7, 5, 9 and 3 haplotypes in *C. parvum* from hosts in 4 different classes (cattle, dogs, humans and birds), and Hap1 and 5 were present in all four groups (Table 1). Hap5 was shared between avian *C. parvum* in white cranes and flamingo in the present study; however, Hap4 was found in only white cranes (Table S3).

The Fst and Nm values based on the 18S rRNA genes between the bird and dog groups were 0.18475 and 1.10318, between the bird and cattle groups were 0.22421 and 0.86503,

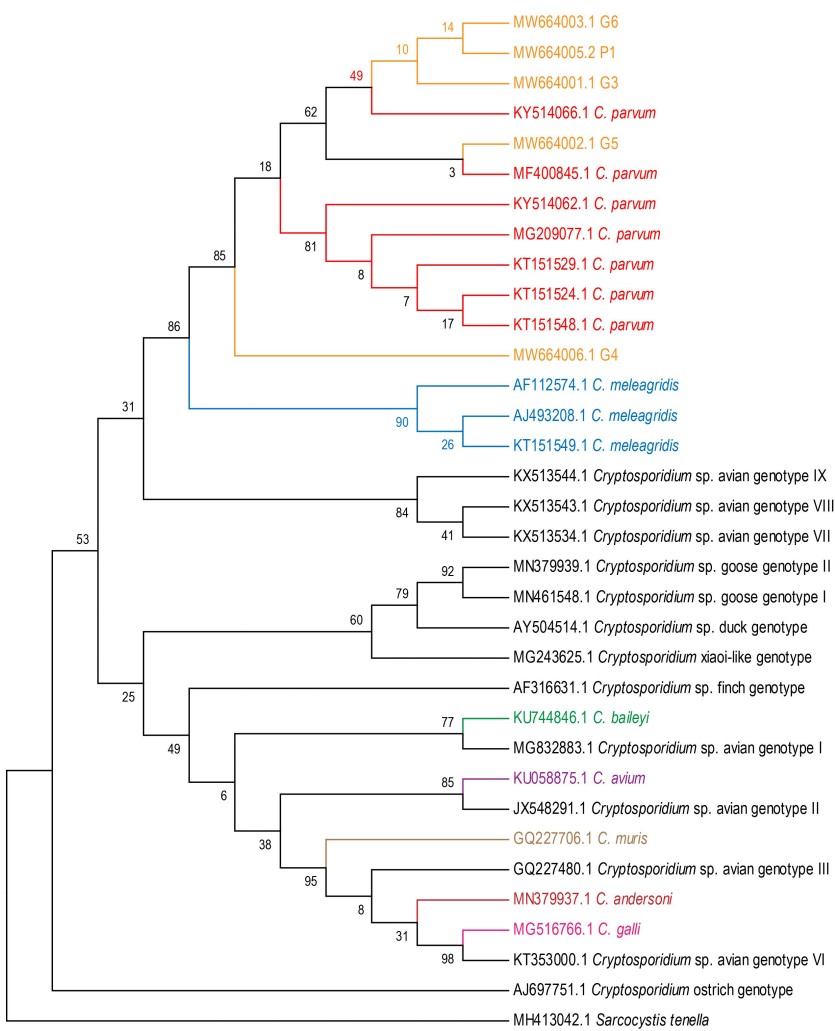

**Figure 1  Phylogenetic tree for *Cryptosporidium* spp.**  Maximum Likelihood tree based on 18S rRNA gene sequences of seven *Cryptosporidium* species (*C. meleagridis*, *C. parvum*, *C. galli*, *C. baileyi*, *C. avium*, *C. muris* and *C. andersoni*) and 13 *Cryptosporidium* genotypes (avian genotypes I–III, and VI–IX; goose genotypes I–II; duck genotype; finch genotype; ostrich genotype; and C.xiaoi-like genotype).

and between the dog and cattle groups were 0.58897 and 0.17447, respectively. The values among *C. parvum* from birds, dogs and cattle indicate gene flow between the *C. parvum* isolates from birds and those from dogs and cattle. The values between the cattle and human groups were 0.05063 and 4.68778, and between the bird and human groups were 0.02137 and 11.4486, respectively, and it seems that birds, like cattle, may transmit *C. parvum* to humans (Table 2).

We calculated the Fst and Nm values of *C. parvum* isolated from three countries: Brazil, Iraq and China (Table S4). The Fst and Nm values between Brazil and China were 0.01530 and 16.08987, between China and Iraq were 0.45714 and 0.29688, and between Brazil and Iraq were 0.24490 and 0.77082, respectively (Table 3). Based on this data, there has been

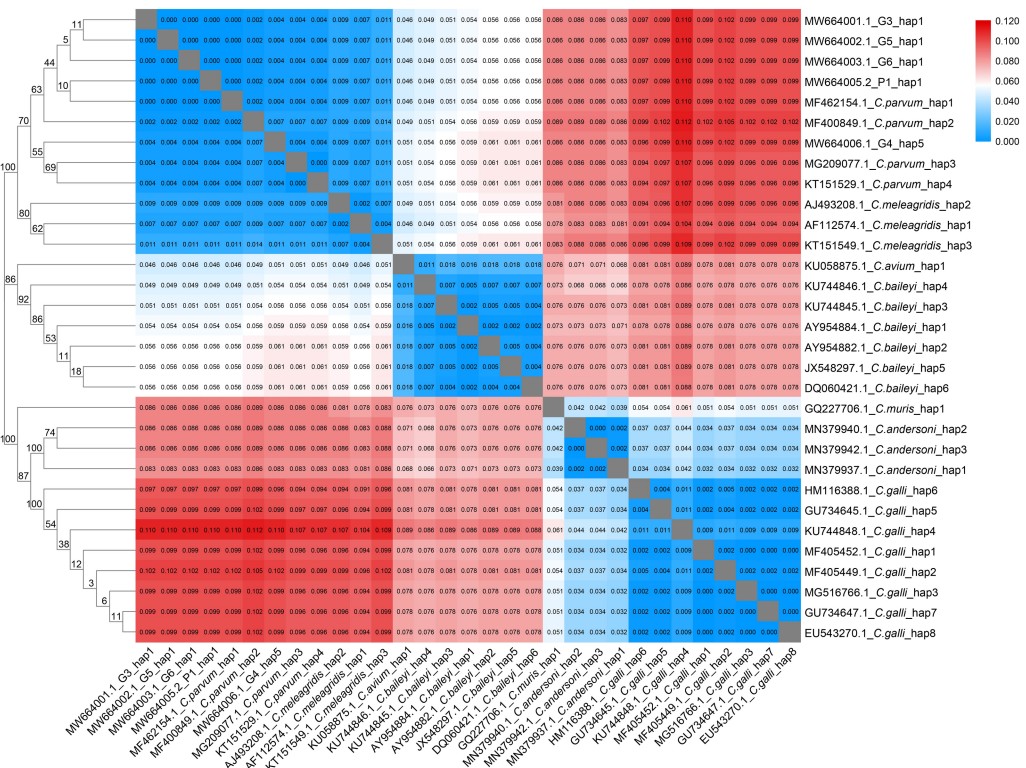

**Figure 2** **Phylogenetic tree and genetic distance among different *Cryptosporidium* haplotype.** Maximum Likelihood tree and genetic distance heatmap based on the haplotypes of the 18S rRNA gene sequences of seven avian *Cryptosporidium* species (*C. parvum*, *C. meleagridis*, *C. avium*, *C. baileyi*, *C. muris*, *C. andersoni* and *C. galli*).

**Table 1** **The genetics parameters of *C. parvum* from four different hosts.**[*]

| Host | N | h | hd | Ps | Pi | Si | $\pi$ |
|------|------|------|--------|------|------|------|---------|
| Cattle | 80 | 7(Hap1-7) | 0.1450 | 11 | 3 | 8 | 0.00177 |
| Dog | 5 | 5(Hap1,5,8-10) | 1.0000 | 16 | 3 | 13 | 0.03318 |
| Human | 61 | 9(Hap1,5,11-17) | 0.5900 | 23 | 19 | 4 | 0.01228 |
| Bird | 20 | 3(Hap1,4-5) | 0.5420 | 3 | 2 | 1 | 0.00571 |
| Total | 166 | 17(Hap1-17) | | | | | |

**Notes.**

    *N *Cryptosporidium* spp. population size.

    h, Number of haplotypes; hd, haplotype diversity; Ps, polymorphic sites; Pi, parsimony informative sites; Si, singleton sites; $\pi$, nucleotide diversity.

    The details of *C. parvum* in different hosts shown in Table S3.

gene flow between *C. parvum* from Brazil and China; however, based on geography, there should theoretically be no gene flow between the two countries.

## DISCUSSION

The rate of *C. parvum* positivity was 1.9% (5/263) in the present study, which was slightly lower than the *C. parvum* positivity rate of 8.5% (58/679) detected in wild birds from

**Table 2    The Fst and Nm values among *C. parvum* from four host groups.**[*]

|  | Cattle | Dog | Human | Bird |
|---|---|---|---|---|
| Cattle | – | 0.17447 | 4.68778 | 0.86503 |
| Dog | 0.58897 | – | 1.89887 | 1.10318 |
| Human | 0.05063 | 0.11634 | – | 11.4486 |
| Bird | 0.22421 | 0.18475 | 0.02137 | – |

**Notes.**
*Fst and Nm values are below and above the diagonal, respectively.

**Table 3    The Fst and Nm values of *C. parvum* among Brazil, Iraq and China.**[*]

|  | Brazil | Iraq | China |
|---|---|---|---|
| Brazil | – | 0.77082 | 16.08987 |
| Iraq | 0.24490 | – | 0.29688 |
| China | 0.01530 | 0.45714 | – |

**Notes.**
*Fst and Nm values are below and above the diagonal, respectively.

Qinghai Lake (*Jian et al., 2021*), and higher than the rate of 0.1% (1/1005) detected in captive pet birds in Henan (*Dong et al., 2021*). Based on these data, the zoo *C. parvum* positivity rate was lower than that in the wild but was higher than that in captive pet birds, but more experiments are needed to support this conclusion.

Currently, we also found that *C. parvum* could infect white crane and flamingo, and we found a unique *C. parvum* haplotype in white crane. The variation in *Cryptosporidium* species plays a key role in the clinical presentation of cryptosporidiosis in humans or animals (*Santin, 2013*); therefore, the *Cryptosporidium* species isolated in this study, *C. parvum*, and its particular haplotype must be studied in greater depth in the future.

In addition, *C. parvum* has been found to infect humans and other mammals (*Ayinmode et al., 2018*; *Moore et al., 2016*). Many of the numerous outbreaks of human cryptosporidiosis reported annually worldwide have been linked to mammals as sources of *C. parvum* infection. The haplotype analysis of *C. parvum* in four different host classes showed that there were two shared haplotypes of *C. parvum* among those hosts (Table 1), which indicated that avian *C. parvum* could spill over into humans and other mammals because of its cross-host transmission ability. The gene flow of *C. parvum* among the four different host classes was also in accordance with the results of share haplotypes (Table 2). Livestock, particularly cattle, are the most important reservoirs of *C. parvum*, potentially posing risks to humans (*Mravcova et al., 2020*). In our study, the Fst value between the bird group and human group was 0.02137, which is even smaller than the value between the cattle group and human group (0.05063). High gene flow suggests that birds, similar to cattle, are crucial reservoir hosts of zoonotic *C. parvum* pathogens, posing risks to humans.

Zoos are important environments for the protection of animals; they are important not only for enhancing the natural resources of animals but also for providing protection and conservation education. As animals in zoos are generally kept in relatively close proximity and have increased opportunities for contact with humans, such as tourists, researchers, feeders, and veterinary staff, sick birds in zoos may potentially be associated with spillover

of zoonotic *Cryptosporidium* pathogens to humans. Therefore, more attention should be given to monitoring for *Cryptosporidium* infection, such as *C. parvum* infection, in birds in zoos due to its potential cross-host transmission ability.

Brazil, a country located in South America, is far from China; the two countries are theoretically geographically isolated, and no natural avian *C. parvum* gene flow should occur between these countries. However, the Fst and Nm results showed higher gene flow between avian *C. parvum* from Brazil and China (Table 3). The reason for this scenario may be that the *C. parvum* pathogen was introduced into the zoo from imported Brazilian flamingos, exchanged animals from other zoos or animals rescued from the wild. This result indicates the need for stricter regulations regarding inspection, quarantine, and isolation protocols for new animal residents.

## CONCLUSIONS

The present study is the first to report that flamingos and white cranes can be infected with *C. parvum*. This knowledge is beneficial for the further protection of white cranes and flamingos and expands the known host range of *C. parvum*. The possibility of *Cryptosporidium* infection is considered in only a small number of wild birds, including those involved in the present study. There are additional potential wild bird hosts that have yet to be identified, and more work on this topic needs to be carried out in the future.

### Funding

This work was supported by the National Natural Science Foundation of China (No. 31970501). The funders had no role in study design, data collection and analysis, decision to publish, or preparation of the manuscript.

### Grant Disclosures

The following grant information was disclosed by the authors:
The National Natural Science Foundation of China: No. 31970501.

### Competing Interests

The authors declare there are no competing interests.

### Author Contributions

- Yaxian Lu conceived and designed the experiments, performed the experiments, analyzed the data, prepared figures and/or tables, authored or reviewed drafts of the paper, and approved the final draft.
- Tianchun Pu conceived and designed the experiments, analyzed the data, authored or reviewed drafts of the paper, and approved the final draft.
- Baohua Ma and Lixin Wang conceived and designed the experiments, analyzed the data, prepared figures and/or tables, authored or reviewed drafts of the paper, and approved the final draft.

- Mengchao Zhou performed the experiments, analyzed the data, authored or reviewed drafts of the paper, and approved the final draft.
- Yu Chen, Xiuyun Li, Changming Zheng, Hetong Liu, Jinpeng Liu, Chunyu Guan, Hongyan Yu, Chunkuo Dai, Yuan Huang and Yuling Yangperformed the experiments, prepared figures and/or tables, and approved the final draft.
- Zhiwei Peng, Lei Han, Hongliang Chai and Zhijun Hou conceived and designed the experiments, authored or reviewed drafts of the paper, and approved the final draft.

### DNA Deposition

The following information was supplied regarding the deposition of DNA sequences:

The Cryptosporidium parvum sequences are available at GenBank: MW664001 to MW664003, MW664005 and MW664006.

### Data Availability

The raw measurements are available in the Supplementary Files.

### Supplemental Information

Supplemental information for this article can be found online at http://dx.doi.org/10.7717/peerj.12825#supplemental-information.

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
