# Peer review of "A survey of Cryptosporidium prevalence among birds in two zoos in China"

_PeerJ, doi:10.7717/peerj.12825_

## Round 0.1 · original submission · Major Revisions

The reviewers have provided a number of comments that, if addressed, would meaningfully improve your manuscript. Please provide a detailed response to each of the comments, explaining how you have changed the manuscript in response to them.

Reviewer 1 ·

Basic reporting

In the present paper, authors collected 263 fecal samples from 43 avian species from zoos. C. parvum was detected by sequencing the 18S rRNA gene of Cryptosporidium. The paper is well organized and generally well written, though there are a few minor errors like grammar and structure throughout the manuscript.

Experimental design

no problems

Validity of the findings

The results demonstrate flamingo and white crane infection by C. parvum for the first time. The results of our study provide an important reference for understanding the host range, biological characteristics, and molecular epidemiology of C. parvum.

Additional comments

Line 40, 45 and 139, it would be better to write the scientific name of the hosts after the common name. I do not find white crane and flamingo in Table S1. There were sever papers that have reported Cryptosporidium in crane. The Table should add common names

Line 50-52, please rewrite “Cryptosporidium is prevalent in fish, amphibians……”.

Line 66-68, C. parvum is Cryptosporidium species, not disease. “…is also prevalent in avian” means what? avian intestine? Please rewrite this sentence.

Line 71-76, “wild birds such as Canada goose, mallard, coot……” this sentence is not complete. Please rewrite.

Line 78-93, the language and structure are so poor. Authors need to rewrite these sentences.

Line 82, does the author have any reference to refer to, if so please add.

Line 95, how did the author collect the samples, please add them.

Line107-117, whether the primer sequence refers to other articles?

Please add Statistical analysis, I think you used "Descriptive analysis (percentages)", as well. So, you should add the sentence.

Please explain, how to protect contamination during the stool collection and collect the stools from the same bird.

Why the author did not perform subtype analysis of positive samples

I find that the sequence uploaded by the author did have several base differences with the standard sequence. Did the author analyzed the cause, whether it could be sequencing problems or other causes of the problem?

Line172-174, how can the author come to this conclusion with few data? please clarify.

Why did the authors choose Brazil and Ireland for data analysis?

Reviewer 2 ·

Basic reporting

The paper presents that the total infection rate of Cryptosporidium in birds in Beijing and Harbin zoos is 1.9%, and it was found that white cranes and flamingos were infected with C. parvum for the first time. By analyzing the gene flow of C. parvum in birds and other animals, it is concluded that bird C. parvum has the risk of spilling to humans and other mammals. However, the paper needs significant improvement before acceptance for publication.

Experimental design

The first discovery of C. parvum infection in white cranes and flamingos and the analysis of gene flow of C. parvum in different hosts are the two highlights of this paper. These research results are conducive to the prevention and control of bird Cryptosporidium in zoos in the future.
The 18S rRNA gene method used in the paper works very well for the prevalence, phylogeny and genetic evolution of Cryptosporidium, this method is a well-established method, on this basis, the gene flow of C. parvum in different hosts was analyzed by biological software.

Validity of the findings

The results of this study are credible, but the description of the results is not detailed enough. For example, at lines 159-167 only list the data but do not make a detailed analysis. It is suggested that you should supplement the description of the results to increase its integrity.

Additional comments

The conclusions presented in this paper are inaccurate and rigorous. For example, it is proposed in the conclusion that C. parvum infection is found in white cranes and flamingos for the first time, but the specific location is not specified. Is this the first time in China or the first time in the world. It is hoped that you will re summarize to draw a correct and unambiguous conclusion.
The English language should be improved to ensure that an international audience can clearly understand your text. It is noted that your manuscript needs careful editing by someone with expertise in technical English editing paying particular attention to English grammar, spelling, and sentence structure so that the goals and results of the study are clear to the reader.
Figures and tables should be annotated in more detail. Whether “Table 5” on the last page should be “Table 3”?

---

## Round 0.2 · accepted · Accept

Thank you for addressing all the comments of the reviewers.